# Corrosion Behavior of Cold-Rolled and Solution-Treated Fe_36_Mn_20_Ni_20_Cr_16_Al_5_Si_3_ HEA in Different Acidic Solutions

**DOI:** 10.3390/ma15207319

**Published:** 2022-10-19

**Authors:** Essam R. I. Mahmoud, Lamiaa Z. Mohamed, Mohamed A. Gepreel, Saad Ebied, Aliaa Abdelfatah

**Affiliations:** 1Department of Mechanical Engineering, Islamic University of Madinah, Madinah 4141, Saudi Arabia; 2Mining, Petroleum and Metallurgical Engineering Department, Faculty of Engineering, Cairo University, Giza 12613, Egypt; 3Materials Science and Engineering Department, Egypt-Japan University of Science and Technology, Alexandria 5221241, Egypt; 4Department of Production Engineering and Mechanical Design, Faculty of Engineering, Tanta University, Tanta 31527, Egypt

**Keywords:** corrosion, high entropy alloys, acidic media, microstructure

## Abstract

New high entropy alloys with good corrosion resistance in severe environment are receiving increasing attention. This work reports upon the microstructure and the corrosion resistance of the non-equiatomic Fe_36_Mn_20_Ni_20_Cr_16_Al_5_Si_3_ alloy in different acidic solutions. This alloy was designed by thermodynamic calculations using CALPHAD SOFTWARE, fabricated through casting, subjected to cold-rolling and solution-treatment, and compared with SS304 stainless steel. The corrosion test was performed through electrochemical behavior in 0.6 M NaCl and 0.6 M NaCl with 0.5 M H_2_SO_4_ and 0.6 M NaCl with 1 M H_2_SO_4_ solutions. Experimental results indicate that the alloy is composed of FCC phase as the main constituent besides a small amount of other BCC/B2 phases and other intermetallics. The corrosion test measurements revealed that cold-rolled Fe_36_Mn_20_Ni_20_Cr_16_Al_5_Si_3_ alloy is more resistant to corrosion in 0.6 M NaCl, while it is more susceptible to localized pits in H_2_SO_4_ to 0.6 M NaCl. Experimental results indicate that the pits are preferentially occurred in the areas of BCC/B2 phase precipitates. The solution-treated Fe_36_Mn_20_Ni_20_Cr_16_Al_5_Si_3_ HEA has the highest corrosion resistance compared to others with the addition of H_2_SO_4_ to 0.6 M NaCl. Surface morphologies of the different conditions were studied, and relevant results were reported.

## 1. Introduction

Developing advanced materials with low cost and good properties operating in severe environments is an urgent demand for many industrial sectors [1]. High entropy alloys (HEAs) are considered one of these promising advanced materials which are solid solution alloys by mixing five or more different elements with equal or near equal atomic ratios. Unlike conventional alloys, HEAs generally appear as simple face, centered cubic bodies, centered cubic or hexagonal close-packed, rather than complex intermetallic phases [2]. This endows HEAs with excellent performance in many industrial applications within a wide range of temperatures from cryogenic to elevated temperatures, for instance, superior mechanical properties, attractive physical properties, superior electrochemical properties, good ductility, high hardness, and high temperature resistance [3]. These distinctive properties make them suitable for various harsh environments such as: thermal, wear, and resistant coatings, chemical plants, components for nuclear power plants, geothermal power plants components, structural materials for transportation, and energy industries, etc. [4]. The corrosion resistance of these HEAs is essential for ensuring the functionality of these alloys to fit in these applications. Accordingly, corrosion behavior of many HEAs have been extensively investigated in various aqueous solutions in recent years. Generally, HEAs gain their good corrosion resistance from the formation of the protective passive film on their surfaces. The nature of the formed passive film plays a critical role on the level of corrosion resistance. Luo et al. [5] studied the corrosion behavior of the Co-Cr-Fe-Mn-Ni HEA in sulfuric acid solution and compared the results with SS304. The HEA shows less corrosion resistance due to the lower content of Cr and the formation of hydroxide in the surface passive film. In another study, Al_x_CrFe_1.5_MnNi_0.5_ HEAs alloys display a better general corrosion resistance than SS304 in 1 M H_2_SO_4,_ but they exhibit a lower pitting corrosion resistance than SS304 in 1 M H_2_SO_4_ and 1 M NaCl. The corrosion resistance decreases with increasing the aluminum content. Aluminum formed a thick porous passive film which enhanced the localized pitting corrosion [6]. Same results were achieved by Niu et al. [7] for the Al_0.5_FeCoCrNiCu HEA, which shows weak corrosion resistance in 1 M NaCl and 0.5 M H_2_SO_4_. Chou et al. [8] reported the effect of mixing sulfates with chlorides on the pitting corrosion of a Co_1.5_CrFeNi_1.5_Ti_0.5_Mo_0.1_ alloy. The pitting potential was critical when the ratio of sulfates to chlorides reached 0.5. Gorr et al. [9] studied the high temperature oxidation properties of the NbMoCrTiAl HEA system, and they reported that the surface of the alloys was covered with a non-protective oxide film when exposed to oxidation test. Quiambao et al. [10] evaluated the passivity of the NiCrFeRuMoW alloy in non-oxidizing sulfate solution at various pH levels. The highly acidic sulfate solution causes spontaneous passivation, and the alkaline solutions cause active-to-passive transition passivation. Chen et al. [11] reported that the Cu_0.5_NiAlCoCrFeSi alloy showed resistance to pitting corrosion in chloride-free solution, and its resistance became less effective in chloride contaminated environments. Qiu et al. [12] reported that CrFeCoNi and CrMnFeCoNi alloys showed pitting resistance to be very close to many austenitic stainless steel alloys. In another work, Qiu et al. [13] studied the corrosion performance of cast Al_x_CoCrFeNiTiy, and reported the formation of oxides of Al_2_O_3_, Cr_2_O_3_, Fe_2_O_3_, and Co_3_O_4_ on the surface after the polarization test of 0.6 M NaCl. Lu et al. [14] reveled that the CoCrFeMnNi system exhibited the obvious characteristics of pitting corrosion in NaCl solution. Zhu et al. [15] studied the effect of potential on the passivity of the surface of the CoCrFeMnNi alloy in a weak alkaline electrolyte solution, and found primary and secondary passivation behavior. Li et al. [16] investigated the corrosion behavior of the AlCoCr-FeNi HEA in 3.5 wt.% NaCl and compared the results with AISI1045 steel. They reported better corrosion resistance of HEA compared to AISI1045 steel, which was attributed to the formation of oxides of Al and Cr on the surface. Moreover, Feng et al. [17] evaluated the corrosion resistance of the CrCoNiN in 3.5 wt.% NaCl solution. The compact passive film improved the corrosion resistance. Another alloy that shows more pitting resistance compared to the SS304 L in 3.5 wt.% NaCl solution is the CoCrFeNi HEA due to its higher Cr content [18]. Besides, Rodriguez et al. [19] evaluated the effect of the different elements on the corrosion resistance of the CoCrFeMnNi HEA system. They found that Cr is the most important element for enhancing corrosion resistance. Cr enhance the corrosion resistance of the Fe_50_Mn_30_Co_10_Cr_10_ alloy against 3.5 wt.% NaCl due to the formation of Cr enrichment passive film [14]. In another study, Torbati-Sarraf et al. [20] investigated the effect of Mn on the pitting corrosion of CrMnFeCoNi and CrFeCoNi systems. They reported that Mn negatively affects the protective ability of the passive film. One of the HEAs families is Fe-Mn-Ni-Cr-Al-Si system. It is composed from relatively low-cost elements, and it contains corrosion resistance elements, such as Cr, whilst Si and Al lower alloy density. Such characteristics made it a promising material for corrosion resistant applications [21].

One of the novel alloys of this system is the Fe_36_Mn_20_Ni_20_Cr_16_Al_5_Si_3_ alloy, which showed very promising mechanical properties and excellent deformability, as reported in [22]. Therefore, studying the corrosion properties in acidic media was an important step towards the complete investigation of the properties of this new HEA.

This current work performs an extensive study on microstructure of the Fe_36_Mn_20_Ni_20_Cr_16_Al_5_Si_3_ HEA in its cold-rolled and solution-treated conditions. An in-depth investigation regarding its corrosion resistance was performed. The effect of different acidic media on Fe_36_Mn_20_Ni_20_Cr_16_Al_5_Si_3_ HEA performance in its cold rolled and solution-treated condition were investigated and compared with the corrosion behavior of stainless steel SS304. Additionally, the surface morphologies of the corroded surfaces were also performed. 

## 2. Materials and Methods

A new non-equiatomic High Entropy Alloy (HEA) system (Fe_36_Mn_20_Ni_20_Cr_16_Al_5_Si_3_) was designed based on the Thermo-Calc software calculations from the following common elements of Fe, Mn, Ni, Cr, Al, and Si to show high strength, good deformability, corrosion resistance, and effective cost. This HEA contains high Mn and Ni content to make FCC, the main phase in the alloy, while Cr is essential to achieve high corrosion-resistance. Al is good in strengthening FCC phase, but it is a BCC stabilizer, so Al-content is limited to only 5%. Si was shown to improve much the corrosion resistance of HEAs, but it is a strong intermetallic former, so it is limited to only 3%. The alloy was produced using an electric arc furnace (ARCAST 200, Maine, ME, USA) under a high purity argon atmosphere. For homogeneity, it was melted 4 times, flipping each time. The ingot of HEA was cut into two parts with rectangular cross-section 10.8 mm by12.66 mm. The two parts were coldrolled to produce two bars with 5 mm diameter that reached about 86% area reduction ratio. One part from the coldrolled HEA was solution-treated at 900 °C for 30 min. The heat treatment temperature was chosen to be relatively low temperature (900 °C) and for a short time (30 min), to allow for recrystallization after cold-rolling but not to allow for grain growth in order to maintain the good properties. Additionally, short solution treatment time was aimed to not make much secondary phases precipitation so as not to affect the corrosion resistance of the alloy. Before the microstructural examination, the cold-rolled, solution-treated HEAs were ground using emery papers up to 1200 grit, then polished by alumina past with 0.3 µm. After that, the electrochemical etching was performed using 10% oxalic acid. The optical microstructures of the cold-rolled and heat-treated Fe_36_Mn_20_Ni_20_Cr_16_Al_5_Si_3_ HEA were examined. The microstructure investigations were carried out using scanning electron microscope (FESEM/QUANTA FEG). X-ray diffraction (D8 Discover with GADDS system, 35 kV, 80 mA) was used in the scanning range of 40 ≤ 2θ ≤ 100° intervals with a step size of 0.05 deg. Their phase identifications were characterized to investigate the crystal structure of the cold-rolled and solution-treated samples. Corrosion samples with 2 mm thickness and 5 mm diameter were cut from both cold-rolled, and solution-treated Fe_36_Mn_20_Ni_20_Cr_16_Al_5_Si_3_ HEA. The investigated samples were ground and polished before the corrosion test. In this work, different corrosion testing medias have been chosen to cover a wide range of working environment severity in order to explore the response and behavior of our new HEA alloy in different working medias, especially the water treatment related conditions. The potentiodynamic-polarization test is performed using three corrosion mediums (0.6 M NaCl, 0.6 M NaCl with 0.5 M H_2_SO_4_ and 0.6 M NaCl with 1.0 M H_2_SO_4_). The target of adding sulfuric acid to sodium chloride is to study the effect of the Cl^−^ and SO_4_^2−^ anions on the investigated HEA as a simulation of the cooling system in the many types of industries. The electrochemical parameters during the test usually include the corrosion potential (E_corr_) and corrosion-current density *(i_corr_*). The parameter, *i_corr_*, can be used to calculate the average corrosion rates from Equation (1) which represent the general corrosion resistance [23,24]:(1)$$\mathrm{Corrosion}\;\mathrm{rate}\;(\mathrm{mm}/\mathrm{year})=3.27\;\times \;10^{-3}\;\times \;\frac{i_{corr}\;}{\rho }\times \;\mathrm{EW}$$
where *ρ* is the density of the alloy (in g/cm^3^), *i_corr_* (in µA/cm^2^) is the corrosion current density, and *EW* is the equivalent weight of the alloy. The calculated *EW* of the investigated HEA is about 20.49 gm (which is derived from the alloy composition, elements atomic weight, and valence).

Surface morphology and chemical analysis of the corroded cold-rolled and heat-treated Fe_36_Mn_20_Ni_20_Cr_16_Al_5_Si_3_ HEA were examined by using scanning electron microscopy (SEM), energy dispersive X-ray analysis (EDX), and mapping for their elemental distribution.

## 3. Results and Discussion

### 3.1. Material Characterizations

The equilibrium phase diagram of the Fe_36_Mn_20_Ni_20_Cr_16_Al_5_Si_3_ HEA predicted by Thermo-Calc calculations [21] is shown in Figure 1. It composed of FCC phase as the main constituent besides a small amount of other phases, such as Sigma, silicides, B2, and other intermetallics. This is due to the higher content of FCC stabilizers (Ni and Mn) and the lower content of BCC stabilizers (Si and Al). SEM micrographs of the cold rolled condition revealed some twinning (green arrows) and slip bands (yellow arrows) as shown in Figure 2. Many intergranular precipitations (red arrows) appeared homogenously at the grain boundaries. Figure 3 showed the SEM micrographs of the solution-treated sample. The deformation features, such as slip bands and twinning, disappeared during the high temperature heat treatment. Many precipitations with different sizes and features appeared as intergranular and transgranular in the matrix. Additionally, the apparent number of precipitates in the annealed condition is significantly increased in comparison to that of the as-rolled condition. The detailed microstructures and the mechanical properties of the alloy in as-rolled and solution-treated conditions were found in our recently published work [22]. Generally, the EDX and mapping of the microstructure in both conditions reveals homogenous distributions of all the alloying elements as shown in Figure 4 and Figure 5. The XRD pattern, as shown in Figure 6, of the cold-rolled condition shows only FCC phase, while its solution-treated sample showed FCC structure with little BCC/B2 phase structure. 

### 3.2. Corrosion Behavior

The corrosion behavior of the Fe_36_Mn_20_Ni_20_Cr_16_Al_5_Si_3_ HEA as cold rolled and solution treated condition in addition to SS304 in different acidic media in terms of potentiodynamic polarization curves are given in Figure 7. Moreover, their electrochemical parameters were obtained though the potentiodynamic polarization tests in solutions of 0.6 M NaCl, 0.6 M NaCl with 0.5 M H_2_SO_4_, and 0.6 M NaCl with 1 M H_2_SO_4,_ and these are listed in Table 1. Generally, the trend of the corrosion current density of the Fe_36_Mn_20_Ni_20_Cr_16_Al_5_Si_3_ HEA in cold-rolled and solution-treated conditions the SS304 samples decreases and then increases. Neither the two HEA in as cold-rolled and solution-treated conditions, nor the SS304 samples established an active to passive transition zone. This may be due to the instantaneous formation of a protective passive film on the surface.

#### 3.2.1. Corrosion Resistance in 0.6 M NaCl

For a solution of only 0.6 M NaCl, the cold-rolled condition has the highest polarization resistance (Rp), which represents the protective ability of the surface passive film, of 33 kohm.cm², which is almost three times that of the solution-treated condition (13.43 kohm.cm²) and four times higher than that of the SS304 (8.77 µA/cm²). This means the oxide passive film that formed on the surface of the cold-rolled HEA condition is stable. An almost reverse trend was observed for the corrosion current density (icorr), which is considered a sign of better corrosion resistance when it shows lower values. The cold-rolled condition has the corrosion current density (icorr) of 12.7 µA/cm² which is almost the value of one over nine times that of the solution-treated condition (75.8 µA/cm²) and almost half of current density of the SS304 (22.4 µA/cm²). According to these results, it can be concluded that the corrosion resistance of the Fe_36_Mn_20_Ni_20_Cr_16_Al_5_Si_3_ HEA in as cold-rolled condition is optimal in 0.6 M NaCl solutions. Furthermore, the annealing treatment reduces the corrosion resistance of the alloy, but still the resistance is much higher compared to SS304. For cold-rolled and solution-treated HEA samples, a slight increase in the current density occurred at 0.1 V, indicating the initiation and growth of stable pitting, while the SS304 sample show a smooth curve (Black curves in Figure 7). Moreover, the polarity of the current density was changed from cathodic to anodic at very close negative corrosion potential values. Figure 8 illustrates the SEM surface morphologies of the Fe_36_Mn_20_Ni_20_Cr_16_Al_5_Si_3_ HEA as cold-rolled and solution-treated conditions after being subjected to a 0.6 M NaCl solution. The samples’ surface appeared to be smooth after being subjected to corrosion test, however, in both conditions, samples have been subjected to varying limited degrees of corrosion. The corrosion occurs in localized areas, especially in the annealed-treated conditions, as shown in Figure 8b. This may be in the areas that sigma, BCC/B2 phases, and other intermetallics were precipitated. Going deeply on the localized corrosion areas, the EDX patterns (Figure 9 and Table 2) of the corroded areas in both cold-rolled and annealed-treated conditions, after being subjected to 0.6 M NaCl, showed relatively higher concentrations of Cr, Si, and Al compared, to the alloy composition, which are considered strong BCC stabilizers, thus increasing the probability of creating these BCC phase in the corroded areas. In addition, the features of the pits appeared very similar to the shape of these precipitates in the SEM images before performing the corrosion test (see Figure 2b). These BCC and sigma phases will deplete the passive film formed (Cr) in this area, which negatively affects the corrosion resistance. Additionally, the content of Al in this alloy is quite high (5 at.%). Al can negatively affect the corrosion resistance of the HEA due to the possibility of formation of thick, porous aluminum oxide film which restricted the formation of the more passive Cr-oxide film, as reported by other works [6]. To confirm this, elemental mapping was performed as shown in Figure 10 and Figure 11, and the results showed that the elements distributed almost uniformly and homogenously after the electrochemical test. This proves that compact passive film was formed homogenously on the surface of the alloys in both conditions even after the corrosion test. In general, the Fe_36_Mn_20_Ni_20_Cr_16_Al_5_Si_3_ HEA in both cold-rolled and solution-treated conditions show a better corrosion resistance than SS304, which has a Cr content of 18% more than that of HEA. This is due to the fact that the Fe_36_Mn_20_Ni_20_Cr_16_Al_5_Si_3_ HEA has both aluminum and chromium. The standard potential of chromium is −0.74 V, which is much higher than that of aluminum (−1.66 V) [25]. In this case, Al can form passive oxide layer on the surface faster than Cr, act as anode in the NaCl solution, and improve the corrosion resistance. 

#### 3.2.2. Corrosion Resistance in in 0.6 M NaCl + 0.5 M H_2_SO_4_

When 0.5 M H_2_SO_4_ was added to 0.6 M NaCl solution, dramatic changes were observed to the corrosion behavior of Fe_36_Mn_20_Ni_20_Cr_16_Al_5_Si_3_ HEA in both conditions. The annealed, treated sample shows the best corrosion resistance in terms of both corrosion current density and polarization resistance. It shows icorr of 178.8 µA/cm², which is almost half of that of the as-rolled condition (309.96 µA/cm²), which is almost three times more resistance than that of as-rolled condition. Moreover, solution treatment condition (5.5 kohm.cm²) has the best corrosion resistance of the cold-rolled (1.7 kohm.cm²) and SS304 sample (0.402 kohm.cm²). Figure 12 show the SEM micrographs of the surface of the Fe_36_Mn_20_Ni_20_Cr_16_Al_5_Si_3_ HEA samples after being subjected to corrosion test in 0.6 M NaCl + 0.5 M H_2_SO_4_ solution. A large amount of localized corrosion pits and products appeared in the as-cold-rolled condition, as shown in Figure 12a. The elements of O, Cr, Fe, and Ni were concentrated in these corrosion areas, as indicated in EDX patterns in Figure 13a. Furthermore, these elements were heterogeneously distributed and concentrated in the corroded area, as shown in EDX patterns in Figure 14 and Table 3. It indicates that some kind of selective dissolution of these elements in the passive film occurred, resulting in few/weak Cr-oxides on the surface, and less efficient corrosion resistance. In addition, the stresses that were generated through the cold-rolling process can affect the corrosion resistance at this severe corrosion resistance. On the other hand, the annealed, treated condition sample shows a relatively smooth surface with small amount of corrosion pits (Figure 12b), and its mapping showed homogenous elemental distributions (Figure 15), indicating superior corrosion resistance at this condition. Generally, the nature of the passive film that formed on the surface, plays an important role in the level of corrosion resistivity. For the cold-rolling sample, a passive film was already there, and protected the surface due to the high content of Cr and Ni. When the sample was subjected to annealed treatment, re-passivation can occur and form a more stable passive film. Although the annealed treatment yields more BCC phases, it gives a chance for surface stress relaxation and the elements to be homogenously distributed in the matrix. This may make the chemical dissolution of the passive film in this severe corrosion condition more difficult resulting in a more compact passive film being formed.

#### 3.2.3. Corrosion Resistance in in 0.6 M NaCl + 1 M H_2_SO_4_

Same trend was obtained when the concentration of H_2_SO_4_ became 1 M. The polarization results of the cold-rolled sample showed a marked increase of corrosion current density to be 563.1 µA/cm² and its corrosion resistance was reduced to a small fraction value of 0.024 kohm.cm². On the contrary, the annealed, treated sample showed a more positive corrosion resistance value, even better than that of lower concentration of H_2_SO_4_. The current density reduced to 101.18 µA/cm² and the resistivity increased to 9.44 µA/cm² indicates superior corrosion resistance in sulfuric acid and sea water. On the other hand, SS304 was in a very bad condition after being subjected to this severe environment. Its corrosion current density jumped to 2715 µA/cm², and its resistivity decrease to lower fraction. These results were confirmed with the SEM micrographs of the corroded surface, as shown in Figure 16, which shows that number of pits, which cover most of the surface, in the as cold-rolled sample is more than that of the solution-treated ones. Additionally, the cold-rolled HEA has a depletion of Cr and Mn elements more than that in the solution-treated HEA, shown in the EDX analysis in Figure 17 and summarized in Table 4. This makes a clear difference in corrosion rate as they are the main elements for passive layer formation. The mapping of the corrosion surface of the as cold-rolled condition, as shown in Figure 18, showed more Cl and O in the pit areas, while there was a depletion of Cr and Mn in these areas. The same trend, but with a lower level, appears in Figure 19 for a solution-treated condition sample. 

## 4. Corrosion Mechanism

In the H_2_SO_4_ solution, the alloying elements affected the corrosion behavior of HEAs. The high concentration of H^+^ influenced the properties of the oxide film, thus affecting the corrosion behavior, and especially the general corrosion resistance. The effect of the Al content on the corrosion behavior of the Al_x_CrFe_1.5_MnNi_0.5_ HEAs in 0.5 M H_2_SO_4_ was investigated [1]. The chloride-containing solution with the absence of Cl^−^ reduces the possibility of the pitting potential [1]. Increasing the Al content, the *E*corr decreased, and *i*corr and *i*pass increased, indicating the reduced general corrosion resistance in the H_2_SO_4_ solution. The adsorptive complexes formed in the acid medium were confirmed as the production of the dissolution of Al in the HEAs by the following mechanism [1]:Al + H_2_O = Al(OH)ad + H^+^ + e^−^(2)
Al(OH)_ad_ + 5H_2_O + H+ = Al^3+^·6H_2_O + 2e^−^(3)

Therefore, the addition of the Al content causes the formation of the porous corrosion product to cover the alloy, and the thickness of the adsorptive layer increases with the amount of Al in the HEA. Moreover, the increased Al content leads to the increased volume fraction of the Al-rich, Cr-depleted BCC phase, whose passive film is more porous and less protective, resulting in the decreased corrosion resistance [1]. Signs of corrosion can be observed in the Al-Ni-rich BCC phase, according to the surface morphology of the alloys after the immersion tests. Iron corrodes in NaCl via its dissolution into ferrous and ferric cations as in Equations (4) and (5) [26,27,28,29,30]. The iron surface develops oxide layers, which slows down and partially protects it from being further attacked by the chloride ions as has been confirmed by the ex and in situ Raman spectroscopy measurements and according to Equations (6) and (7). The passive film is more compact when FeO is present at higher concentrations, thus increasing its corrosion resistance [31]. The Fe_x_O_y_ constituted the final dense rust near the inner layer under the simulated seawater environment. Equilibrium equations for the stable range of Fe_3_O_4_ during the corrosion process were listed in Equations (8)–(10):Fe = Fe^2+^ + 2e^−^(4)
Fe^2+^ = Fe^3+^ + e^−^(5)
Fe + ½ O_2_ + H_2_O = Fe(OH)_2_(6)
3Fe(OH)_2_ +½ O_2_ = Fe_3_O_4_ +3H_2_O(7)
3Fe_2_+ + 4H_2_O = Fe_3_O_4_ + 8H^+^ + 2e^−^(8)
3FeO + H_2_O = Fe_3_O_4_ + 2H^+^ + 2e^−^(9)
2Fe_3_O_4_ + H_2_O = 3Fe_2_O_3_ + 2H^+^ + 2e^−^(10)

The formation of Cr_2_O_3_ prevents Fe from oxidation as shown in Equation (11). Cr concentration enhances HEAs’ capacity to passivate in aqueous solutions including chloride and sulfuric acid, which increases the passive coating’s durability and pitting corrosion resistance. On the other side, there were excessive Cr results in severe pitting corrosion because of Cr segregation [32].
2Cr + 3H_2_O → Cr_2_O_3_ + 6H^+^ + 6e^−^(11)

Mn can be used as a low-cost replacement component for alloys in mildly corrosive aqueous solutions, but it has a negative impact on HEA corrosion in sulfuric acid and chloride-containing aqueous solutions [21]. Mn might form Mn-rich compounds (oxides and hydroxides) during the whole corrosion process as illustrated in Equations (12)–(15) [14,15,16,17,18].
Mn^2+^ + 2FeOOH = MnFe_2_O_4_ + 2H^+^(12)
2Fe_3_O_4_ + 3Mn^2+^ + 4H_2_O = 3MnFe_2_O_4_ + 8H^+^ + 2e^−^(13)
3Mn(OH)^3−^ + 2Fe_3_O_4_ + H^+^ = 3MnFe_2_O_4_ + 5H_2_O + 2e^−^(14)
3MnFe_2_O_4_ + 4H_2_O = 6FeOOH + Mn_3_O_4_ + 2H^+^ +2e^−^(15)

Ni is in a metallic form at the metal/oxide junction, which adds to the decline in dissolution [32]. NiO and Ni(OH)_2_ are possible, as indicated in Equations (16) and (17). Elemental segregation makes alloys more prone to pitting and diminishes their capacity to passivate. On the other hand, the high entropy impact of HEAs promotes the uniform distribution of components needed by the permeation hypothesis, increasing HEAs’ capacity for passivation. As a result, a compact and uniform passive coating has a propensity to form on the surface of HEAs. It is important to note that the high entropy effect is not the only factor contributing to the superior protection of passive films produced on HEAs [32].
Ni + H_2_O → Ni(OH)_2_ + 2H^+^ + 2e^−^(16)
Ni(OH)_2_ → NiO + 2H^+^ + 2e^−^(17)

The corrosion behavior of the developed HEA greatly depends on the concentrations and type of the corrosion medium. In 0.6 M NaCl, the corrosion produces pits through the passive layer formed. With the addition of H_2_SO_4_ solution to 0.6 M NaCl, this led to a breakdown of the passive film due to increasing the diffusivity of the chloride and sulfide ions through microcracks [21,33].

The high concentration of Cr in the developed HEA may be advantageously predicted for the material’s resistance to corrosion. The increased Ni content may also reduce the total rates at which Fe and Cr dissolve. As the passive film generated in 1.0M H_2_SO_4_ is very unstable, the obtained HEA has less corrosion resistance than SS304 [5,21].

In cooling water, the sulfuric acid is added with NaCl to increase the activity of chloride ions and prevent the precipitation. The ability of the Cl^−^ and SO_4_^2−^ anions to adsorb on the passive film and create an electrostatic field across the film/electrolyte interface may be related to pitting corrosion. When the field reaches a certain value, the adsorbed anions penetrate the oxide film, particularly at flaws and defects. When the penetrated Cl^−^ and SO_4_^2−^ anions reach the metal surface, they promote local anodic dissolution, which results in the formation of a pit nucleus. Following this, pit growth occurs rapidly as a result of an increase in corrosive ion concentration caused by migration, increasing the acidity within the pits [34].

Figure 20 indicates the difference between corrosion rates of cold-rolled HEA, solution-treated HEA, and SS304 in different corrosion medium conditions. For cold-rolled Fe_36_Mn_20_Ni_20_Cr_16_Al_5_Si_3_ HEA in 0.6 M NaCl with different concentrations of H_2_SO_4_, the corrosion rate increases due to degradation of Cr and Mn. These elements are responsible for the formation of the passive layer. Otherwise, the corrosion rate in 0.6 M NaCl is decreased due to formation of the passive layer from Cr and Mn oxide, which appeared from EDX analysis of the corroded HEA in different conditions. Solution treatment of investigated HEA was responsible for increasing the corrosion resistance of it. It may be due to homogenization of the microstructure, new phase formation (BCC/B2 phase structure), and the stress relived, as shown in Figure 1 and Figure 4. Generally, the corrosion behavior of the different conditions of developed HEA is better than corrosion behavior of SS304 in the examined corrosion mediums as shown Figure 20. 

## 5. Conclusions

This newly designed Fe_36_Mn_20_Ni_20_Cr_16_Al_5_Si_3_ high entropy alloy is studied in cold-rolled and solution-annealed at 900 °C for 30 min conditions. The microstructure and corrosion behavior of cold-rolled and heat-treated Fe_36_Mn_20_Ni_20_Cr_16_Al_5_Si_3_ HEA compared with SS304 was investigated in 0.6 M NaCl, 0.6 M NaCl with 0.5 M H_2_SO_4_, and 0.6 M NaCl with 1 M H_2_SO_4_ solutions. The major results are summarized as the following:The Fe_36_Mn_20_Ni_20_Cr_16_Al_5_Si_3_ alloy was composed mainly of FCC phase in the cold-rolled condition in addition to few amounts of the BCC/B2 phase and other intermetallics that were precipitated at the grain boundaries after subsequent solution annealing.In 0.6 M NaCl solution, the corrosion resistance of the cold-rolled Fe_36_Mn_20_Ni_20_Cr_16_Al_5_Si_3_ HEA is higher compared with that of the solution annealed condition and the SS304 alloy.The addition of H_2_SO_4_ to the 0.6 M NaCl deplete the Cr and Mn of the as cold-rolled condition and decrease the Fe_36_Mn_20_Ni_20_Cr_16_Al_5_Si_3_ alloy corrosion resistance. Under these conditions, the solution annealed Fe_36_Mn_20_Ni_20_Cr_16_Al_5_Si_3_ HEA showed the best corrosion resistance.The surface passive films provide the protection for the underlying HEA from further dissolution which improve the corrosion resistance in H_2_SO_4_ solution.

## Figures and Tables

**Figure 1 materials-15-07319-f001:**
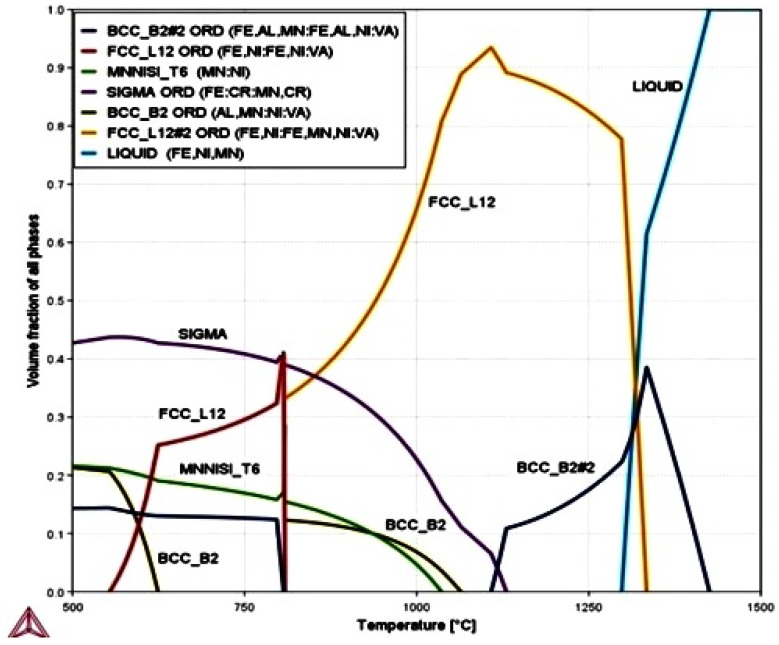
Equilibrium phase diagram for Fe_36_Mn_20_Ni_20_Cr_16_Al_5_Si_3_ HEA generated by thermodynamic calculations using CALPHAD SOFTWARE [21].

**Figure 2 materials-15-07319-f002:**
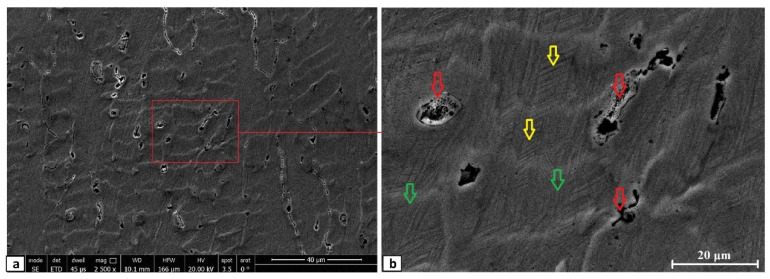
Different magnified SEM micrographs of cold rolled Fe_36_Mn_20_Ni_20_Cr_16_Al_5_Si_3_ high-entropy alloy, where (**b**) is the zoomed-in area of (**a**). (Green arrows shows twinning and yellow arrows shows slip bands).

**Figure 3 materials-15-07319-f003:**
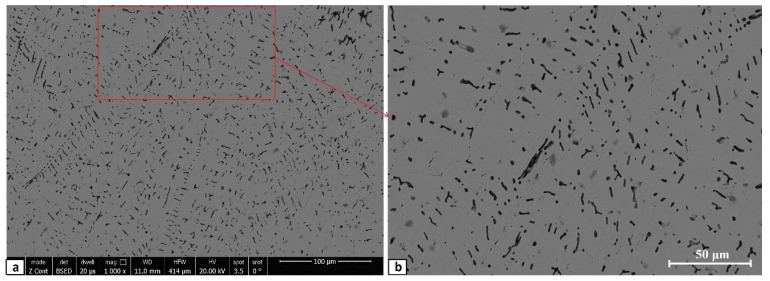
Different magnified SEM micrographs of solution-treated Fe_36_Mn_20_Ni_20_Cr_16_Al_5_Si_3_ high-entropy alloy, where (**b**) is the zoomed-in area of (**a**).

**Figure 4 materials-15-07319-f004:**
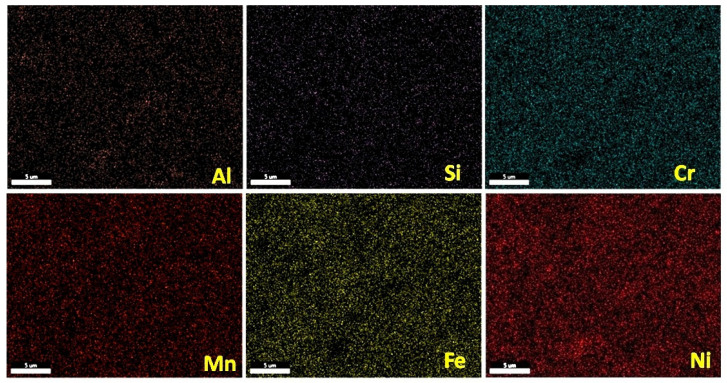
Mapping of cold-rolled Fe_36_Mn_20_Ni_20_Cr_16_Al_5_Si_3_ high-entropy alloy.

**Figure 5 materials-15-07319-f005:**
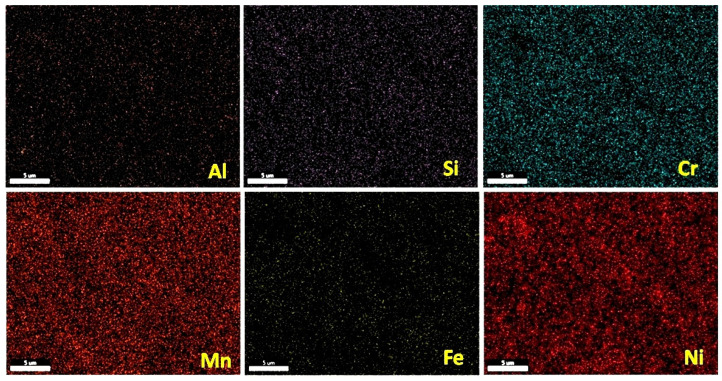
Mapping of solution-treated Fe_36_Mn_20_Ni_20_Cr_16_Al_5_Si_3_ high-entropy alloy.

**Figure 6 materials-15-07319-f006:**
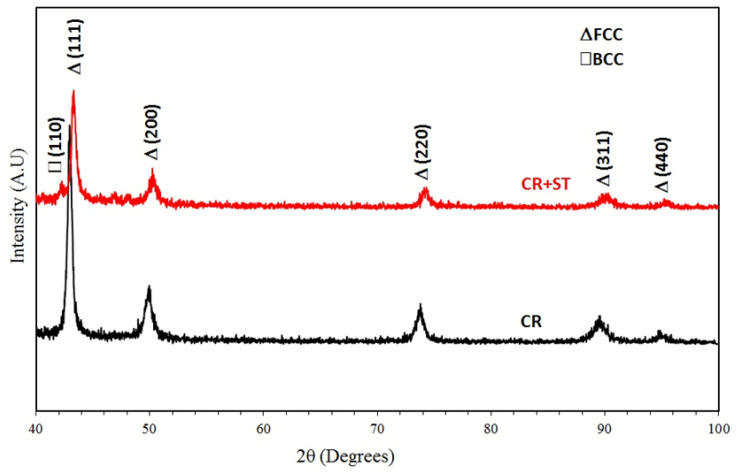
XRD patterns of the cold rolled (CR) and solution-treated (CR + ST) Fe_36_Mn_20_Ni_20_Cr_16_Al_5_Si_3_ HEA [23].

**Figure 7 materials-15-07319-f007:**
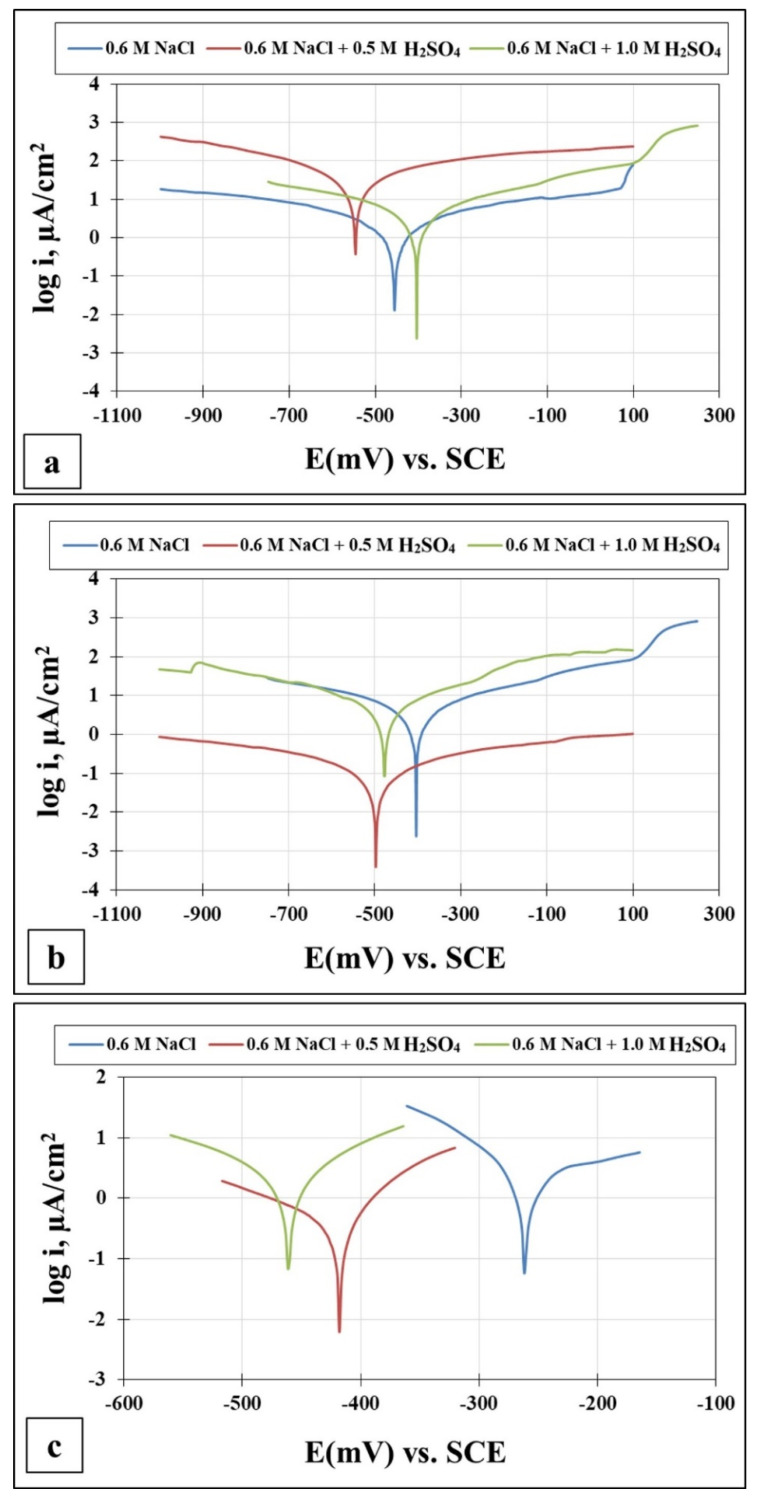
The corrosion behavior of the (**a**) cold-rolled HEA, (**b**) solution-treated HEA and (**c**) SS304 in different acidic medias.

**Figure 8 materials-15-07319-f008:**
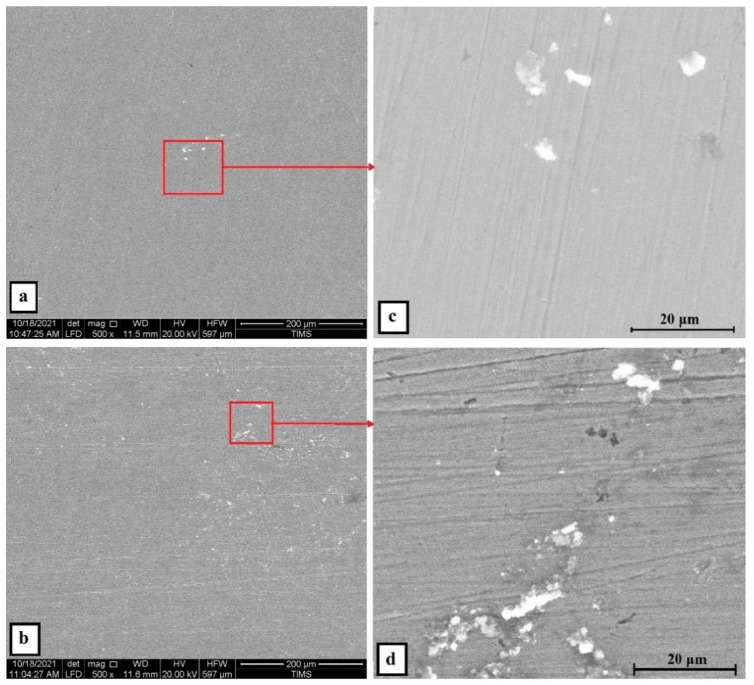
SEM surface morphologies of the Fe_36_Mn_20_Ni_20_Cr_16_Al_5_Si_3_ HEA (**a**,**c**) cold-rolled, and (**b**,**d**) solution-treated conditions after subjected to 0.6 M NaCl solution.

**Figure 9 materials-15-07319-f009:**
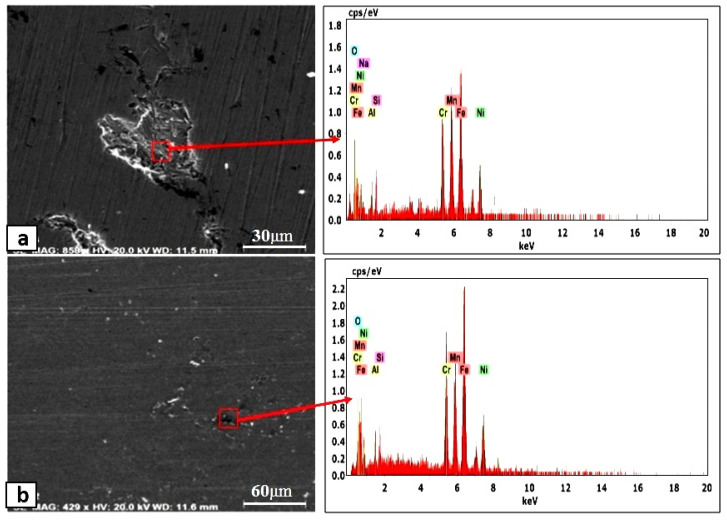
The EDX patterns of the corroded Fe_36_Mn_20_Ni_20_Cr_16_Al_5_Si_3_ HEA in (**a**) cold-rolled and (**b**) solution-treated conditions in 0.6 M NaCl.

**Figure 10 materials-15-07319-f010:**
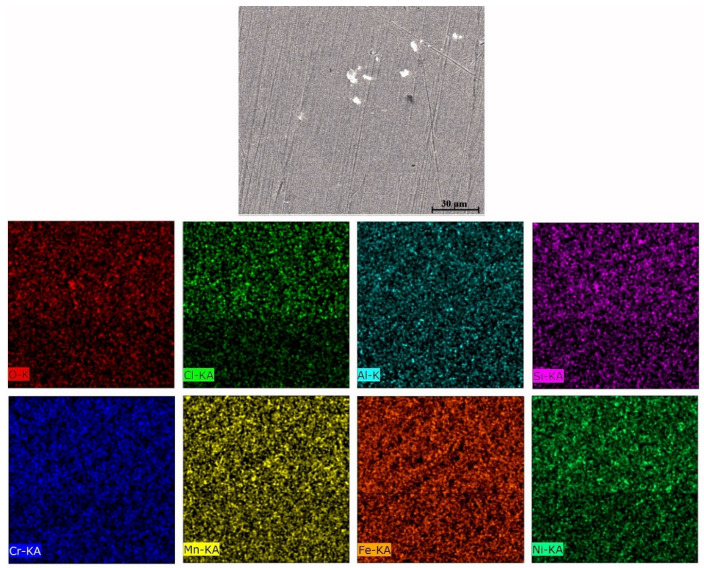
EDX elemental mapping of the corroded cold-rolled Fe_36_Mn_20_Ni_20_Cr_16_Al_5_Si_3_ HEA in 0.6 M NaCl.

**Figure 11 materials-15-07319-f011:**
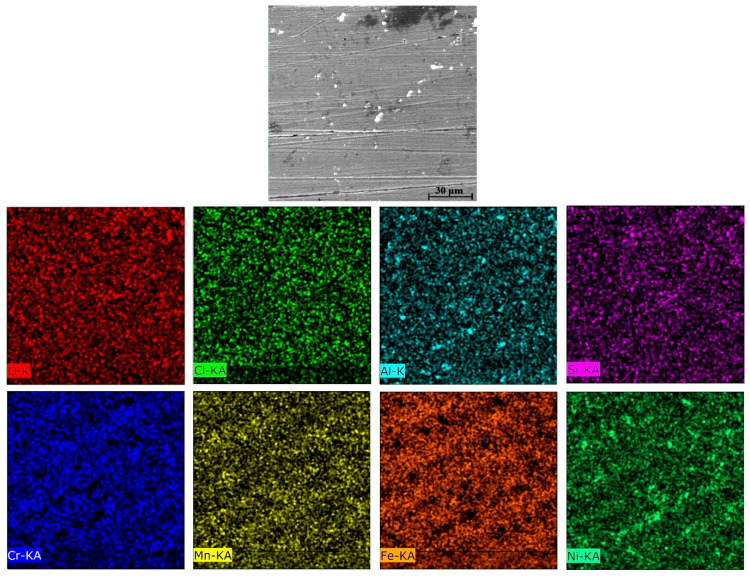
EDX elemental mapping of the corroded solution-treated Fe_36_Mn_20_Ni_20_Cr_16_Al_5_Si_3_ HEA in 0.6 M NaCl.

**Figure 12 materials-15-07319-f012:**
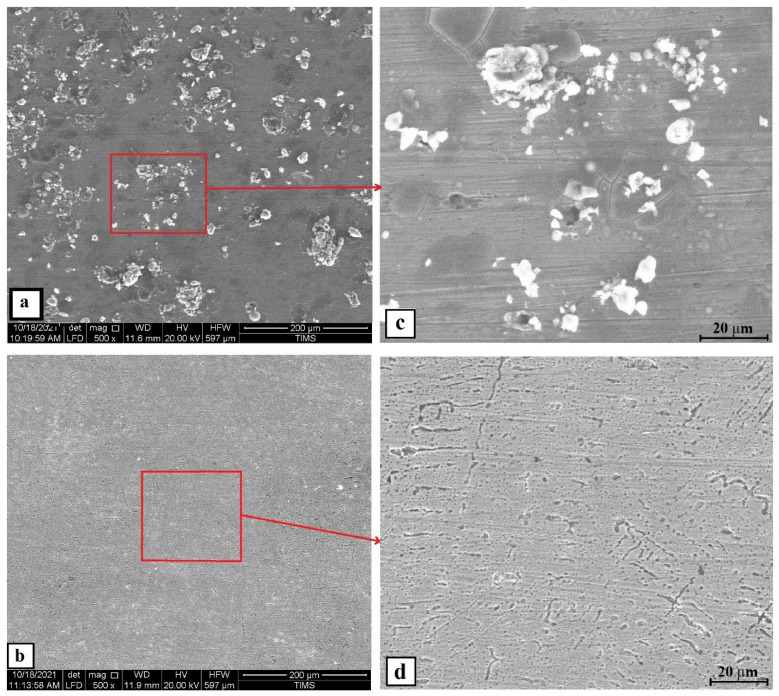
The SEM surface morphologies of the Fe_36_Mn_20_Ni_20_Cr_16_Al_5_Si_3_ HEA as (**a**,**c**) cold-rolled and (**b**,**d**) solution-treated condition after subjected to 0.6 M NaCl + 0.5 M H_2_SO_4_ solution.

**Figure 13 materials-15-07319-f013:**
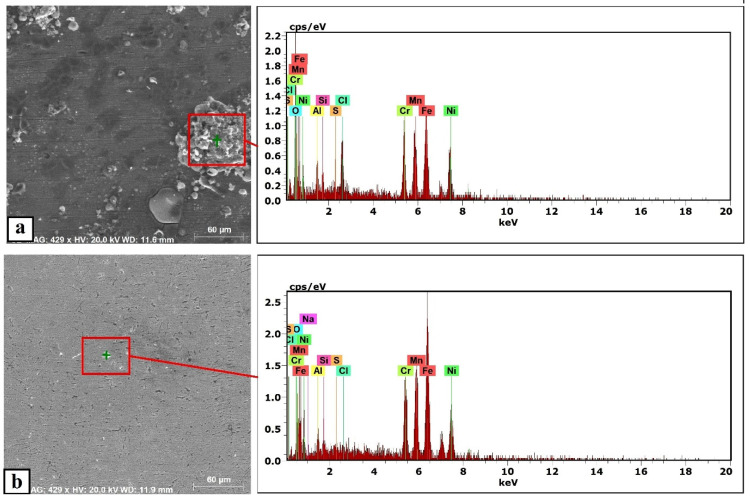
EDX patterns of the corroded HEA in (**a**) cold-rolled and (**b**) solution-treated condition in 0.6 M NaCl + 0.5 M H_2_SO_4_ solution.

**Figure 14 materials-15-07319-f014:**
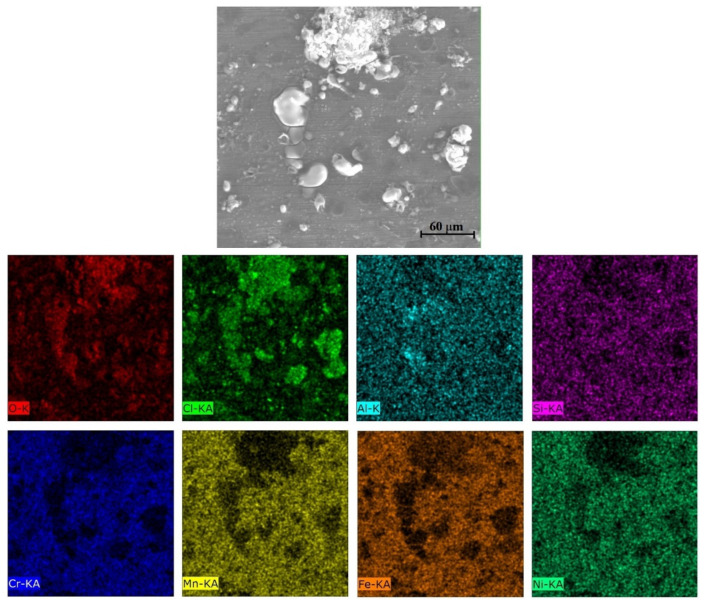
EDX elemental mapping of the corroded cold-rolled Fe_36_Mn_20_Ni_20_Cr_16_Al_5_Si_3_ HEA in 0.6 M NaCl with 0.5 M H_2_SO_4_.

**Figure 15 materials-15-07319-f015:**
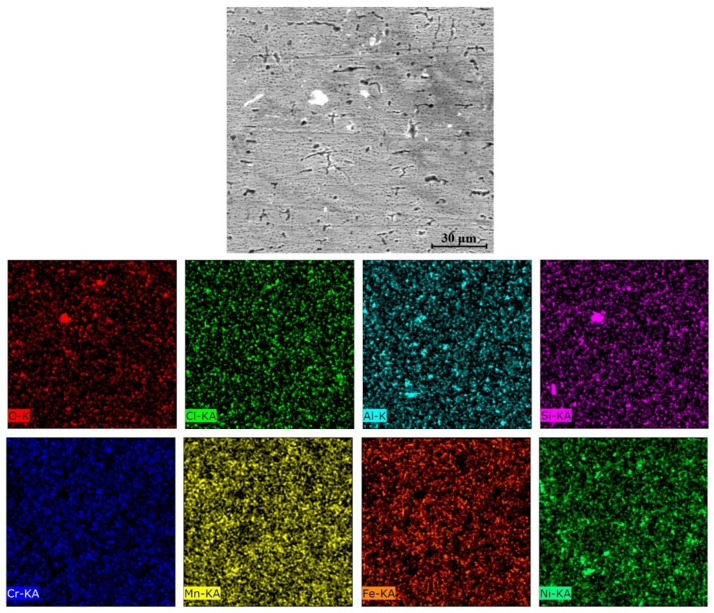
EDX elemental mapping of the corroded solution-treated (CR + ST) Fe_36_Mn_20_Ni_20_Cr_16_Al_5_Si_3_HEA in 0.6 M NaCl with 0.5 M H_2_SO_4_.

**Figure 16 materials-15-07319-f016:**
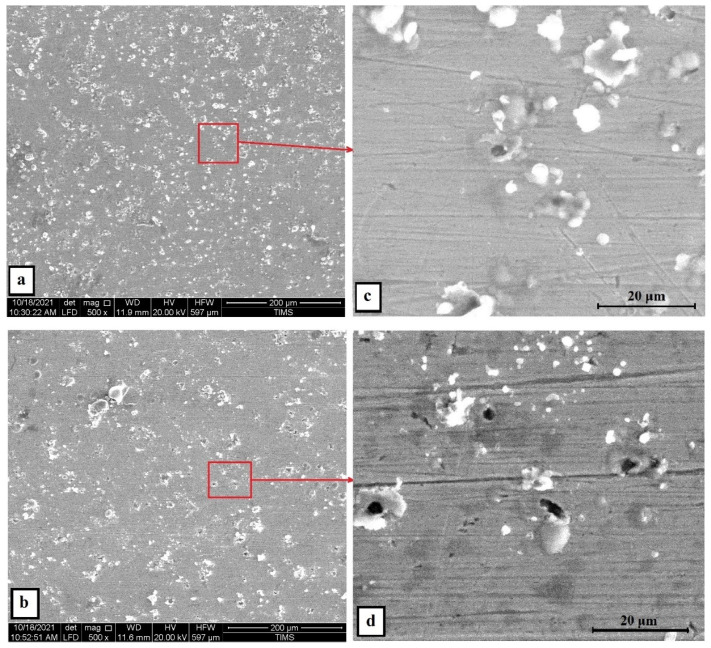
The SEM surface morphologies of the Fe_36_Mn_20_Ni_20_Cr_16_Al_5_Si_3_ HEA (**a**) as cold-rolled and (**b**) solution-treated condition after subjected to 0.6 M NaCl + 1 M H_2_SO_4_ solution, (**c**) the zoomed-in area of (**a**), and (**d**) the zoomed-in area of (**b**).

**Figure 17 materials-15-07319-f017:**
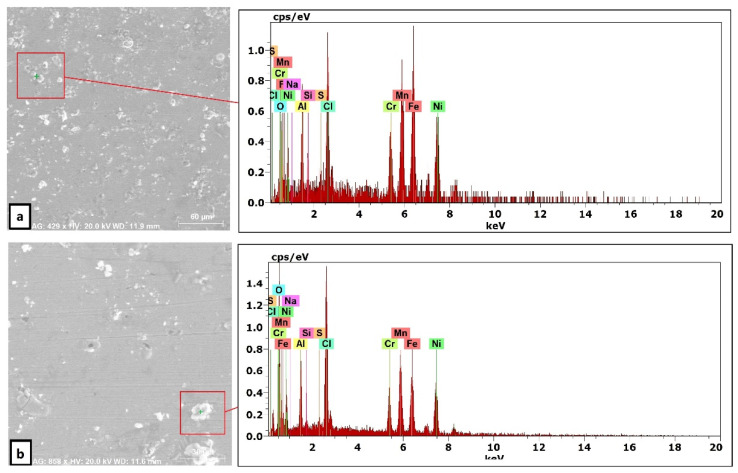
EDX patterns of the corroded HEA in cold-rolled and solution-treated condition in 0.6 M NaCl + 1 M H_2_SO_4_ solution, where (**a**) SEM image of one of small pits, and (**b**) SEM image of one of large pits.

**Figure 18 materials-15-07319-f018:**
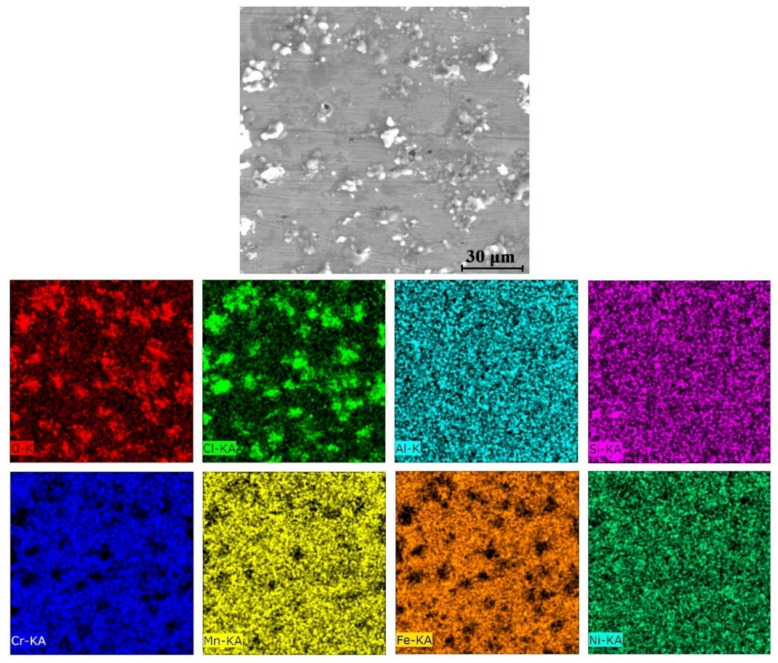
EDX elemental mapping of the corroded cold-rolled Fe_36_Mn_20_Ni_20_Cr_16_Al_5_Si_3_ HEA in 0.6 M NaCl with 1 M H_2_SO_4_.

**Figure 19 materials-15-07319-f019:**
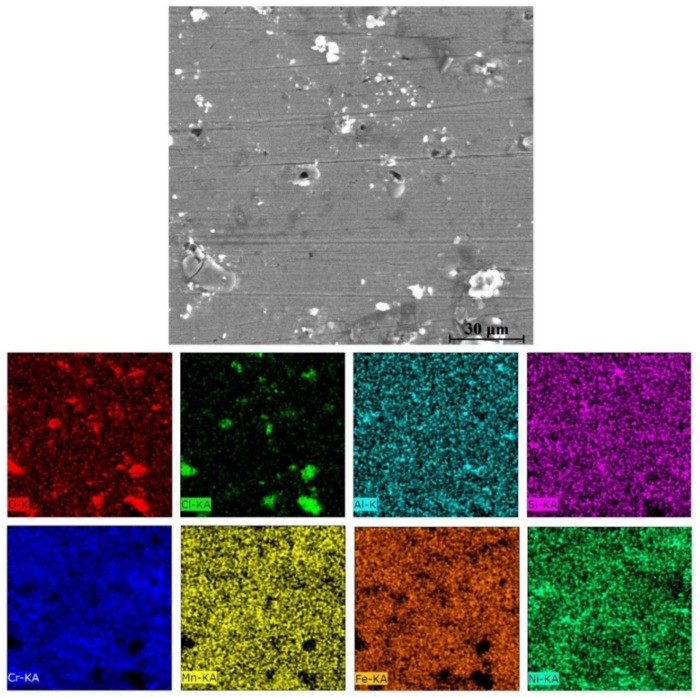
EDX elemental mapping of the corroded solution-treated Fe_36_Mn_20_Ni_20_Cr_16_Al_5_Si_3_ HEA in 0.6 M NaCl with 1 M H_2_SO_4_.

**Figure 20 materials-15-07319-f020:**
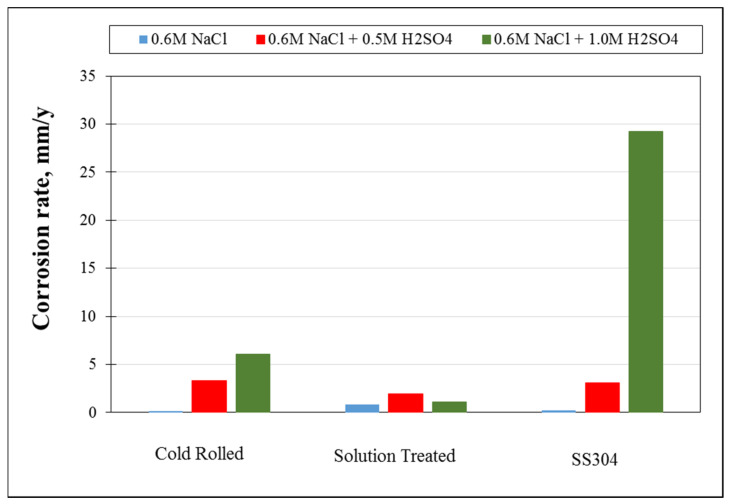
Comparison between the corrosion rates of cold-rolled and solution-treated Fe_36_Mn_20_Ni_20_Cr_16_Al_5_Si_3_ HEA, and SS304 in different corrosion medium conditions.

**Table 1 materials-15-07319-t001:** The electrochemical parameters of cold-rolled HEA, solution-treated HEA, and SS304.

Conditions		Ecorr, mV	icorr, µA/cm²	Rp, kohm.cm²	CR, mm/y
Cold-rolled	0.6 M NaCl	−455.1	12.70	33.00	0.136
	0.6 M NaCl + 0.5 M H_2_SO_4_	−546.1	309.96	1.70	3.33
	0.6 M NaCl + 1 M H_2_SO_4_	−417.8	563.10	0.024	6.056
Solution-treated	0.6 M NaCl	−403.8	75.85	13.43	0.815
	0.6 M NaCl + 0.5 M H_2_SO_4_	−497.1	178.8	5.57	1.92
	0.6 M NaCl + 1 M H_2_SO_4_	−477.1	101.18	9.44	1.095
SS304	0.6 M NaCl	−261.2	22.4	8.77	0.241
	0.6 M NaCl + 0.5 M H_2_SO_4_	−417.8	288.2	0.402	3.10
	0.6 M NaCl + 1 M H_2_SO_4_	−460.9	2715	0.095	29.20

**Table 2 materials-15-07319-t002:** EDX analysis of the corrosion products on the surface of cold-rolled and solution-treated Fe_36_Mn_20_Ni_20_Cr_16_Al_5_Si_3_ HEA in 0.6 M NaCl.

Conditions	Elements, at.%									
	O	Na	Cl	S	Al	Si	Cr	Mn	Fe	Ni
Cold-rolled HEA	12.6	2.4	0.0	0.0	4.0	3.8	16.0	18.1	31.1	12.0
Solution-treated HEA	2.1	0.0	0.0	0.0	4.7	2.8	19.0	15.8	39.5	16.3

**Table 3 materials-15-07319-t003:** EDX analysis of the corrosion products of the surface on cold-rolled (CR) and solution-treated (CR + ST) HEA in 0.6 M NaCl with 0.5 M H_2_SO_4_.

Conditions	Elements, at.%									
	O	Na	Cl	S	Al	Si	Cr	Mn	Fe	Ni
Cold-rolled HEA	50.1	0.0	3.8	0.1	3.3	1.4	7.4	9.0	15.1	9.8
Solution-treated HEA	20.3	5.0	0.5	0.9	6.6	3.1	9.7	14.4	25.0	14.5

**Table 4 materials-15-07319-t004:** EDX analysis of the corrosion products on the surface of cold-rolled and solution-treated Fe_36_Mn_20_Ni_20_Cr_16_Al_5_Si_3_ HEA in 0.6 M NaCl with 1.0 M H_2_SO_4_.

Conditions	Elements, at.%									
	O	Na	Cl	S	Al	Si	Cr	Mn	Fe	Ni
Cold-rolled HEA	56.5	0.0	9.5	0.0	6.0	1.2	2.1	5.9	6.1	12.7
Solution-treated HEA	52.0	0.2	8.3	0.1	3.8	1.0	5.7	8.1	9.7	10.9

## Data Availability

Not applicable.

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
