# Peer review of "Corrosion Behavior of Cold-Rolled and Solution-Treated Fe_36_Mn_20_Ni_20_Cr_16_Al_5_Si_3_ HEA in Different Acidic Solutions"

_materials, 2022, doi:10.3390/ma15207319_

Round 1
Reviewer 1 Report
All the reviewer comments are attached in the file.

Author Response
Dear Prof. Reviewer
We thank you very much for your concern about our paper (Ref. No.: materials-1867457) titled “Corrosion behavior of cold rolled and solution treated Fe36Mn20Ni20Cr16Al5Si3 HEA in different acidic solutions" and appreciate your valuable comments on our work. Kindly, please find our reply on your comments in the following points.
Comment #1: “Developing advanced materials with low cost and good properties for severe environments is an urgent demand for many industrial sectors [1].” have HEAs low cost than stainless steels (e.g., 304 and 316 L)?.
Authors Reply
We mean low-cost alloy system in high entropy alloys category, as it consists mainly of the common low-cost elements (such as Fe, Al, Mn, Ni, Si). However, this low-cost HEAs may not be cheaper than the commercial stainless steels 304 and 316L alloys from price per kg of alloys point of view. Even though, the newly developed low-cost HEAs may be cheaper than commercial stainless steels in terms of specific properties (such as specific strength, easy production, comparable corrosion resistance and so on).
Comment #2: “Unlike conventional alloys, HEAs generally appeared as simple face centered cubic, body centered cubic or hexagonal close packed rather than complex intermetallic phases [2]”. The majority of conventional alloys have simple crystal structure. Clarify this sentence.
Authors Reply
Due to the sluggish diffusion nature of HEAs, the formations of complex and intermetallic brittle phases are kinetically not favorable -in most cases- although it may be thermodynamically possible. This gives good room for easy production and processing of HEAs while gaining many improved properties in them due to the new compositional range compared to conventional alloys.
Comment #3: The authors performed an adequate literature review about the HEAs corrosion resistance but not indicated the main drawbacks, microstructure aspects, and reaction mechanisms that explain the inferior corrosion resistance of HEAs in operational mediums. In addition, some references indicated the inferior corrosion HEA in relation to commercial workhorse alloys (e.g., SS304). Also, why the composition modification used by the authors can improve the HEAs corrosion resistance? Thus, discuss how the inferior corrosion behavior can limit or even barrier the applications of HEa and how to overcome these limitations. In addition, the manuscript's purpose is not clearly evident, which difficult the manuscript understood.
Authors Reply
This work is a part of project aiming to design relatively low-density high entropy alloys and study their properties. One of our outputs is Fe36Mn20Ni20Cr16Al5Si3 alloy as reported in our newly published paper [22]. This new Fe36Mn20Ni20Cr16Al5Si3 alloy showed very promising mechanical properties and excellent deformability as reported in [23]. Therefore, studying the corrosion properties in acidic media was an important step towards complete investigation of the properties of this new HEA, which is the scope of this paper.
Comment #4: Fe36Mn20Ni20Cr16Al5Si3. This alloy was previously studied, or this is an inedita study? if previously studied show the references and main results. If not, what are the criteria for alloy composition selection? This is the critical point of the manuscript, which not explain the reasons to select something.
Authors Reply
The Fe36Mn20Ni20Cr16Al5Si3 is new alloy recently designed by our team. It was chosen from many other compositions based on their thermodynamic calculations, phase diagram prediction by ThermoCalc software, microstructure and mechanical properties. as reported in [22]. This new Fe36Mn20Ni20Cr16Al5Si3 alloy showed very promising mechanical properties and excellent deformability as reported in [23].
Comment #5: The parameters of heat treatment were based in some previous work or in authors trial to error experiments? The authors must explaining why make this selection.
Authors Reply
From the previous experience and literature, as well as the phase diagram of the alloy, the heat treatment condition was chosen to be relatively low temperature (900 oC) and for short time (30 min.), to allow for recrystallization after cold rolling but not to allow for grain growth in order to maintain the good properties. Also, short solution treatment time was aimed not to make much secondary phases precipitation so as not to deteriorate the corrosion resistance of the alloy.
This explanation was inserted in the modified manuscript.
Comment #6: Some previous surface modification was applied to corroded surface or the samples, after potentiodynamic-polization, was insert directly on SEM chamber? Better explain the experimental procedure to analyze the corroded samples in SEM.
Authors Reply
The corrosion test was explained in detail in the submitted modified manuscript.
Comment #7: Compare the thermo-calc results with the literature simulations performed for FeMnNiCraLSi systems to verify if the model assumptions really represent the alloy's physical metallurgy.
Authors Reply
Due to the sluggish diffusion nature of HEAs, the Thermo-Calc software was successful to high extent in the prediction of the main formed phases in FeMnNiCrAlSi HEAs during the rapid solidification process (such as production by arc melting in Cu-cooled crucible process) as reported by the present authors in [23].
Comment #8: Please showed in all micrographs the microstructural aspects indicated in the manuscript text.
Authors Reply
Done in the submitted manuscript.
Comment #9: This abrupt variation in volume fraction (Figure 1) rises from computational instability; so, refine this date, taking off the abrupt variation about 800°C.
Authors Reply
Done in the submitted manuscript.
Comment #10: Explain the selection of corrosion medium concentrations and put this in material and methods.
Authors Reply
Done in the modified manuscript
Comment #11: Add a figure comparing the corrosion behaviors between cold rolled, solution HT, and SS304 for a worse corrosion medium condition.
Authors Reply
Done in modified manuscript. Figure 20 was added to indicates the difference between corrosion rates of cold rolled HEA, solution treated HEA, and SS304 in different corrosion medium conditions.
Fig. 20. Comparison between the corrosion rates of cold rolled HEA, solution treated HEA, and SS304 in different corrosion medium conditions.
Comment #12: Increase the SEM image resolution (referents to corrosion analyzed conditions) and also pointed the morphological and microstructural aspects indicated in the manuscript text.
Authors Reply
Done in the modified manuscript for all figures.
Comment #13: In item 4, the authors take numerous references and make hypotheses about the corrosion mechanism but do not show results that confirm the statements. So, reinforce the discussion about this item. Also, make a critical comparison between the tested corrosion mediums and the behavior of HEA alloy (both rolled and solution treated), clarifying why each condition has the behavior verified experimentally. Finally, conclude if the developed HEA is better/worse than the SS 304. The manuscripts need the author's opinion, not only show the results.
Authors Reply
Section 4 was modified according to the reviewer comments.
Comment #14: The authors numerous times justify the corrosion behavior by alloy segregation; however, does not have results to support this statement. Please, perform some solidification simulation (e.g., Scheil–Gulliver model) or analytical data to reinforce the solidification segregation mechanism of HEA. In addition, perform EDS mapping in as-rolled and solution-treated conditions without corrosion products to confirm the simulation tendency.
Authors Reply
EDX mapping was performed on samples before corrosion tests and the results were shown in Figures 4 and 5. It shows normal homogenous distribution of the alloying elements.
Comment #15: “The addition of H2SO4 to the 0.6M NaCl deplete the Cr and Mn of the as cold rolled condition and decrease the Fe36Mn20Ni20Cr16Al5Si3 alloy corrosion resistance. Under these conditions, the solution annealed treated Fe36Mn20Ni20Cr16Al5Si3 HEA showed the best corrosion resistance”. how the corrosion medium modified the alloy chemical composition?.
Authors Reply
The corrosion behavior of the developed HEA strongly depends on the concentrations and type of the corrosion medium. In 0.6M NaCl, the corrosion produces pits through the passive layer formed. Where the addition of H2SO4 solution to 0.6M NaCl led to breakdown of the passive film due to increasing the diffusivity of the chloride and sulfide ions through microcracks [23].
Comment #16: As the manuscript objective was not clear, the item 5 was not conclusive. So, first clearly defines the manuscript objectives and posterity make the conclusions.
Authors Reply
The objective of this work is: The newly developed low-cost FeMnNiCrAlSi HEA based on the showed excellent cold workability and heat treatability and very promising mechanical properties compared to the commercial st.st. therefore, the corrosion resistance of this new alloy is investigated in this work compared to the commercial st.st. alloys in order to open the room for replacing it in some harsh environment applications.
Thanks for your guidance and I hope to cooperate with you in the near future.
Best Regards

Reviewer 2 Report
The paper needs careful revision. I prefer rejection.
1. The authors should do EBSD to find out the change of microstructure after cold rolling and solution treatment.
2. XPS should be conducted. The scale formed on the surface after corrosion tests should also be examined.
3. The corrosion mechanism proposed by the authors is well known. No new finding has been given. Also the corrosion mechanisms proposed in the paper is lack of evidence.
4. There are some techniques being used, and the manuscript looks like a scientific report rather than a scientific paper. Most of them were only described in phenomenon. They should be discussed in more depth by correlating each other.
Author Response
Dear Prof. Reviewer
We thank you very much for your concern about our paper (Ref. No.: materials-1867457) titled “Corrosion behavior of cold rolled and solution treated Fe36Mn20Ni20Cr16Al5Si3 HEA in different acidic solutions" and appreciate your valuable comments on our work. Kindly, please find our reply on your comments in the following points.
Comment #1: The authors should do EBSD to find out the change of microstructure after cold rolling and solution treatment.
Authors Reply
We agree that the texture in the alloys may be resulted due to cold rolling or solution annealing. However, texturing study using EBSD is out of the scope of this paper (only corrosion performance) and will be presented in a future work. In addition, the mechanical properties of this Fe36Mn20Ni20Cr16Al5Si3 new high entropy alloy in its cold rolling and solution treatment conditions were investigated in detail in our recent published work [23]. By the way, we added in the modified submitted manuscript the EDX mapping of the elements of both conditions before corrosion test. The main scope of the paper is studying the corrosion resistance of new Fe36Mn20Ni20Cr16Al5Si3 alloy.
[23] Mahmoud, E.R.I.; Shaharoun, A.; Gepreel M. A.; Ebied, S. Studying the Effect of Cold Rolling and Heat Treatment on the Microstructure and Mechanical Properties of the Fe36Mn20Ni20Cr16Al5Si3 High Entropy Alloy. Entropy 2022, 24, 1040. https://doi.org/10.3390/e24081040
Comment #2: XPS should be conducted. The scale formed on the surface after corrosion tests should also be examined
Authors Reply
We agree that XPS may give further information about the structure of the corrosion products. However, due to lake of XPS facility, we performed SEM with EDX analysis on the corroded areas. Also, EDX mapping was performed also in the same areas. We think those tests are enough to evaluate the corroded surface after corrosion test, and gave us enough information about the corrosion products, as discussed in the paper.
Comment #3 The corrosion mechanism proposed by the authors is well known. No new finding has been given. Also the corrosion mechanisms proposed in the paper is lack of evidence.
Authors Reply
This Fe36Mn20Ni20Cr16Al5Si3 alloy under study in this paper is new alloy showed very promising mechanical properties and excellent deformability. Therefore, studying the corrosion properties in acidic media was an important step towards complete investigation of the properties of this new HEA. Regardless the proven corrosion mechanism is unique or not, it was very important to know it for further consideration in future development of HEAs with high corrosion resistance.
Comment #4: There are some techniques being used, and the manuscript looks like a scientific report rather than a scientific paper. Most of them were only described in phenomenon. They should be discussed in more depth by correlating each other.
Authors Reply
With respect to the reviewer comment, the paper investigated deeply the corrosion resistance of the Fe36Mn20Ni20Cr16Al5Si3 high entropy alloy in both rolled and solution treated conditions in different chloride and sulfide media. Section #4 (Corrosion Mechanism) was modified in the submitted modified manuscript according to the reviewer comment. The work can be extended for investigations by experts in future in the field that may extract more information and give more recommendations.
Thanks for your guidance and I hope to cooperate with you in the near future.
Best Regards

Reviewer 3 Report
1. Authors have to improve the editorial work on the manuscript. As an examples, in the first statement in line 92 a verb is missing and it seems that this sentence should be in a previous paragraph. Such editorial issues are in the whole manuscript and they should be corrected.
2. Authors should put more focus in explaining why they chose Fe-Mn-Ni-Cr-Al-Si system for the investigation.
3. Figure 2 - more detailed characterization of the microstructure should be taken. It is difficult to distinguish if the black areas are precipitates or holes/pores. Authors should do EDS analysis to confirm the chemical composition of the precipitates. These analysis should be also done in Figure 3.
4. Potentiodynamic polarization tests presented in Figure 5, 6 and 7 should have been taken to higher potential, because then probably for each state a breakage potential would occur.
5. Figure 8 - the scale bars are not visible.
6. Figure 9 - the quality of the images should be higher, because scale bars are hardly seen.
7. Figure 10 - the quality of the EDS mapping is poor and it should be improved. Actually, the scan was stopped after reaching a half, therefore lower part of each map is darker.
8. Scale bars in Figure 12 have to be improved.
9. In Figure 14 the EDS maps have to be enlarged and also the description which element is on each map has to be enlarged because it is difficult to read.
10. Scale bars in Figure 16 have to be enlarged.
11. In Figure 18 and Figure 19 the EDS maps have to be enlarged and also the description which element is on each map has to be enlarged because it is difficult to read. Also, the maps should be taken in a better quality.
Author Response
Dear Prof. Reviewer
We thank you very much for your concern about our paper (Ref. No.: materials-1867457) titled “Corrosion behavior of cold rolled and solution treated Fe36Mn20Ni20Cr16Al5Si3 HEA in different acidic solutions" and appreciate your valuable comments on our work. Kindly, please find our reply on your comments in the following points.
Comment #1: Authors have to improve the editorial work on the manuscript. As an examples, in the first statement in line 92 a verb is missing and it seems that this sentence should be in a previous paragraph. Such editorial issues are in the whole manuscript and they should be corrected.
Authors Reply
The whole paper was carefully revised by an English native reviewer and the necessary corrections were done.
Comment #2: Authors should put more focus in explaining why they chose Fe-Mn-Ni-Cr-Al-Si system for the investigation.
Authors Reply
This Fe36Mn20Ni20Cr16Al5Si3 alloy was designed based on thermodynamic calculations and phase diagram prediction by ThermoCalc software, as reported in Ref. [22]. This new Fe36Mn20Ni20Cr16Al5Si3 alloy showed very promising mechanical properties and excellent deformability as reported in Ref. [23]. Therefore, studying the corrosion properties in acidic media was an important step towards complete investigation of the properties of this new HEA, the scope of this paper.
Comment #3 Figure 2 - more detailed characterization of the microstructure should be taken. It is difficult to distinguish if the black areas are precipitates or holes/pores. Authors should do EDS analysis to confirm the chemical composition of the precipitates. These analysis should be also done in Figure 3.
Authors Reply
The part of microstructure was detailed investigated in our recently published paper [23].
Comment #4: Potentiodynamic polarization tests presented in Figure 5, 6 and 7 should have been taken to higher potential, because then probably for each state a breakage potential would occur.
Authors Reply
That is true, it might be break at high potential as in literature at 1.1 V [35]. The authors expected that, and it was written inside the manuscript as the following:
The corrosion behavior of the developed HEA strongly depends on the concentrations and type of the corrosion medium. In 0.6M NaCl, the corrosion produces pits through the passive layer formed. Where the addition of H2SO4 solution to 0.6M NaCl led to breakdown of the passive film due to increasing the diffusivity of the chloride and sulfide ions through microcracks [23].
[35] Li, W.; Guo, W.; Zhang, H.; Xu, H.; Chen, L.; Zeng, J.; Liu, B.; Ding, Z. Influence of Mo on the Microstructure and Corrosion Behavior of Laser Cladding FeCoCrNi High-Entropy Alloy Coatings. Entropy 2022; 24, 539.
Comment #5: Figure 8 - the scale bars are not visible.
Authors Reply
New figure with higher resolution was inserted in the modified manuscript
Comment #6: Figure 9 - the quality of the images should be higher, because scale bars are hardly seen.
Authors Reply
New figure with higher resolution was inserted in the modified manuscript
Comment #7: Figure 10 - the quality of the EDS mapping is poor and it should be improved. Actually, the scan was stopped after reaching a half, therefore lower part of each map is darker
Authors Reply
New figure with higher resolution was inserted in the modified manuscript
Comment #8: Scale bars in Figure 12 have to be improved
Authors Reply
New figure with higher resolution was inserted in the modified manuscript
Comment #9: In Figure 14 the EDS maps have to be enlarged and also the description which element is on each map has to be enlarged because it is difficult to read.
Authors Reply
New figure with higher resolution was inserted in the modified manuscript
Comment #10: Scale bars in Figure 16 have to be enlarged.
Authors Reply
New figure with higher resolution was inserted in the modified manuscript
Comment #11: In Figure 18 and Figure 19 the EDS maps have to be enlarged and also the description which element is on each map has to be enlarged because it is difficult to read. Also, the maps should be taken in a better quality.
Authors Reply
New figure with higher resolution was inserted in the modified manuscript
Thanks for your guidance and I hope to cooperate with you in the near future.
Best Regards

Reviewer 4 Report
The topic addressed in this work is considered of interest for the potential readers. Different corrosive scenarios are tested, and the results are deeply analysed. The draft structure and research soundness is correct, and the introduction is elaborated. Nevertheless, according the reviewer point of view, some mandatory improvements must be done prior to publication.
Abstract/Figure 1
Substitute Thermo-Calc calculations by a more general term that reflects the essence of the calculation procedure. Why we should rely in these calculations? Are the results and the conclusions of the research not valid if other software is used for the same purpose? (Obviously not, but this must be clear from the Abstract and the Materials and Methods section).
Materials and Methods
Explain the strategy and fundamentals of the calculation performed by Thermo-Calc
Results and discussion
Please, make sure that the provided polarization curves of figure 7 have enough resolution, in the current draft is difficult to read the legend of the graphs.
The authors hypothesize about the influence of residual stresses, introduced during cold working and relieved during annealing, in the corrosion behaviour. From the reviewer point of view, it is not clear from the draft the role of the residual stress (as in the performed experimental tests the microstructural changes cannot be dissociated from the stress relief, both due to the annealing stage). It must be explained in detail the expected influence of residual stress on the pitting resistance. Please, provide measurements or estimations of the surface residual stress due to cold rolling. Moreover, it is suggested to perform corrosion tests on the cold rolled samples after stress relief by mechanical methods (milling, for example) without annealing treatment, to isolate the stress effect on the corrosion resistance (preliminary pitting resistance tests will clarify this in a very approachable way).
Conclusions
The mechanical properties and cold workability of the HEA are referenced but not included/commented within this draft. Therefore, these properties cannot be included in the conclusions section, or these HEA properties must be included and commented within the text.
Author Response
Dear Prof. Reviewer
We thank you very much for your concern about our paper (Ref. No.: materials-1867457) titled “Corrosion behavior of cold rolled and solution treated Fe36Mn20Ni20Cr16Al5Si3 HEA in different acidic solutions" and appreciate your valuable comments on our work. Kindly, please find our reply on your comments in the following points.
Comment #1: The topic addressed in this work is considered of interest for the potential readers. Different corrosive scenarios are tested, and the results are deeply analysed. The draft structure and research soundness is correct, and the introduction is elaborated. Nevertheless, according to the reviewer point of view, some mandatory improvements must be done prior to publication.
Authors Reply
We appreciate the reviewer comment and understanding that our paper concerns the corrosion behavior of one of the novel high entropy alloys; Fe36Mn20Ni20Cr16Al5Si3 HEA in its cold rolled and solution-treated conditions. This alloy was chosen from different others because it showed very promising mechanical properties and excellent deformability. A brief summary was given about the microstructure of the alloy and its phases through SEM, EDS and XRD analysis. The other points: alloy design, microstructure and mechanical properties are presented recently in different published papers in MDPI journals. The corrosion tests were performed using three corrosion mediums (0.6M NaCl, 0.6M NaCl with 0.5M H2SO4 and 0.6M NaCl with 1.0M H2SO4). The corroded surfaces were examined through SEM and EDS mapping. The corrosion mechanism of this novel promising alloy was discussed in a systematic and professional way. The corrosion results presented in this paper is also promising since the corrosion resistance is competing and even better than the commercial stainless steels as reference alloy (besides the already know promising deformability and mechanical properties of the same alloy).
Comment #2: Abstract/Figure 1
Substitute Thermo-Calc calculations by a more general term that reflects the essence of the calculation procedure. Why we should rely in these calculations? Are the results and the conclusions of the research not valid if other software is used for the same purpose? (Obviously not, but this must be clear from the Abstract and the Materials and Methods section).
Authors Reply
We agree with the reviewer comment that there are different methods and software to predict the phase stability and help in designing new alloys with targeted microstructure based on the thermodynamic parameter change as a function of composition. However, the Thermo-Calc software is the tool we have, and it showed very good results to predict the formed phases in such high entropy alloy system (as reported in recent work Ref. 22-23). Out of the works, this present alloy showed a promising mechanical properties and excellent cold workability that encouraged us to extend investigating this alloy through corrosion behavior in different medium, the scope of this paper. Therefore, and based on the reviewer advice, we substitute the name of the software “Thermo-Calc” with a general term “thermodynamic calculations using CALPHAD SOFTWARE”.
22) Mahmoud, E.R.I.; Shaharoun, A.; Gepreel M. A.; Ebied, S. Phase Prediction, Microstructure and Mechanical Properties of Fe–Mn–Ni–Cr–Al–Si High Entropy Alloys. Metals 2022, 12, 1164. https://doi.org/10.3390/met12071164
23) Mahmoud, E.R.I.; Shaharoun, A.; Gepreel M. A.; Ebied, S. Studying the Effect of Cold Rolling and Heat Treatment on the Microstructure and Mechanical Properties of the Fe36Mn20Ni20Cr16Al5Si3 High Entropy Alloy. Entropy 2022, 24, 1040. https://doi.org/10.3390/e24081040
Comment #3 Materials and Methods
Explain the strategy and fundamentals of the calculation performed by Thermo-Calc
Authors Reply
The present alloy is selected mong different alloys composition. The high entropy alloy of targeted high corrosion resistance and at the same time shows good cold workability is aimed to be of FCC main phase with minors BCC solid solutions without the formation of intermetallic. Therefore, to make the idea more clear, the following part was introduced in the materials and method section:
“This HEA contains high Mn and Ni content to make FCC is the main phase in the alloy, while Cr is essential to achieve high corrosion resistance. Al is good in strengthening FCC phase but it is BCC stabilizer, so Al-content is limited to only 5%. Si was proved to improve much the corrosion resistance of HEAs but it is strong intermetallic former, so, it is limited to only 3%.”
Comment #4: Results and discussion
Please, make sure that the provided polarization curves of figure 7 have enough resolution, in the current draft is difficult to read the legend of the graphs.
Authors Reply
New figure with higher resolution was inserted in the modified manuscript
Comment #5:
The authors hypothesize about the influence of residual stresses, introduced during cold working and relieved during annealing, in the corrosion behaviour. From the reviewer point of view, it is not clear from the draft the role of the residual stress (as in the performed experimental tests the microstructural changes cannot be dissociated from the stress relief, both due to the annealing stage). It must be explained in detail the expected influence of residual stress on the pitting resistance. Please, provide measurements or estimations of the surface residual stress due to cold rolling. Moreover, it is suggested to perform corrosion tests on the cold rolled samples after stress relief by mechanical methods (milling, for example) without annealing treatment, to isolate the stress effect on the corrosion resistance (preliminary pitting resistance tests will clarify this in a very approachable way).
Authors Reply
The amount and type of stress (compressive or tensile) appear to be important factors in pit nucleation [1]. The surface finish caused by residual stress had a significant impact on the polarization curve. With increasing compressive stress, the critical current density for passivation and the passive current density decreased rapidly. Because the passive film can be produced and maintained at low current density, the presence of compressive residual stress makes producing the passivation film easier regardless of the surface condition varied by surface finish. This could explain why the reduction in interatomic spacing caused by compressive stress at the surface can aid in the formation and maintenance of the passivation film. Not only does compressive residual stress improve mechanical properties, but it also improves corrosion resistance [2]. Heavy cold working, which induces compressive stresses, is associated with a recovery in the stoichiometry of the protective passive layer and resistance to pit nucleation.
[1] G. Monrrabal, A. Bautista, S. Guzman, C. Gutierrez, F. Velasco, Influence of the cold working induced martensite on the electrochemical behavior of AISI 304 stainless steel surfaces, journal of materials research and technology, 8(1), 1335-1346, 2019
[2] O. Takakuwa, H. Soyama, Effect of Residual Stress on the Corrosion Behavior of Austenitic Stainless Steel, Advances in Chemical Engineering and Science, 2015, 5, 62-71
Comment #6: Conclusions
The mechanical properties and cold workability of the HEA are referenced but not included/commented within this draft. Therefore, these properties cannot be included in the conclusions section, or these HEA properties must be included and commented within the text.
Authors Reply
The sentence that refers to the mechanical properties of the alloy was removed, and the conclusion part was modified according to the reviewer comment.
Thanks for your guidance and I hope to cooperate with you in the near future.
Best Regards

Round 2
Reviewer 1 Report
The authors solved and answer all the reviewer comments.
Author Response
Dear Prof. Reviewer
We thank you very much for your concern about our paper (Ref. No.: materials-1867457) titled “Corrosion behavior of cold rolled and solution treated Fe36Mn20Ni20Cr16Al5Si3 HEA in different acidic solutions" and appreciate your valuable comments on our work. Kindly, please find our reply on your comments in the following points.
Comment #1: The authors solved and answer all the reviewer comments.
Authors Reply
We appreciate the reviewer comment and thank him for his comment. The whole paper was reviewed again to improve the quality of the work.
future.
Best Regards

Reviewer 2 Report
The authors acclaimed that “This Fe36Mn20Ni20Cr16Al5Si3 alloy under study in this paper is new alloy showed very promising mechanical properties and excellent deformability. Therefore, studying the corrosion properties in acidic media was an important step towards complete investigation of the properties of this new HEA. Regardless the proven corrosion mechanism is unique or not, it was very important to know it for further consideration in future development of HEAs with high corrosion resistance”. I can't agree with this. If so, lots of papers can be prepared in a very short time with different alloys. Therefore, it is true the reviewers are not satisfied with reading this kind of shallow and superficial paper.
Also the authors said "texturing study using EBSD is out of the scope of this paper (only corrosion performance)". The corrosion behavior of alloys is strongly affacted by the microstructure. With the necessary microstructure charactrzation, how can the authors explain the reasons for the change of corrosion resisttance caused by cold rolling?
In a word, I have to reject this paper because there are not any good creative points the readers are very interesting.
Author Response
Dear Prof. Reviewer
We thank you very much for your concern about our paper (Ref. No.: materials-1867457) titled “Corrosion behavior of cold rolled and solution treated Fe36Mn20Ni20Cr16Al5Si3 HEA in different acidic solutions" and appreciate your valuable comments on our work. Kindly, please find our reply on your comments in the following points.
Comment #1: The authors acclaimed that “This Fe36Mn20Ni20Cr16Al5Si3 alloy under study in this paper is new alloy showed very promising mechanical properties and excellent deformability. Therefore, studying the corrosion properties in acidic media was an important step towards complete investigation of the properties of this new HEA. Regardless the proven corrosion mechanism is unique or not, it was very important to know it for further consideration in future development of HEAs with high corrosion resistance”. I can't agree with this. If so, lots of papers can be prepared in a very short time with different alloys. Therefore, it is true the reviewers are not satisfied with reading this kind of shallow and superficial paper.
Also the authors said "texturing study using EBSD is out of the scope of this paper (only corrosion performance)". The corrosion behavior of alloys is strongly affacted by the microstructure. With the necessary microstructure charactrzation, how can the authors explain the reasons for the change of corrosion resisttance caused by cold rolling?
Authors Reply
With respect to the reviewer comment, our paper concerns the corrosion behavior of one of the novel high entropy alloys; Fe36Mn20Ni20Cr16Al5Si3 HEA in its cold rolled and solution-treated conditions. This alloy was chosen from different others because it showed very promising mechanical properties and excellent deformability. A brief summary was given about the microstructure of the alloy and its phases through SEM, EDS and XRD analysis. The other points: alloy design, microstructure and mechanical properties are presented recently in different published papers in MDPI journals. The corrosion tests were performed using three corrosion mediums (0.6M NaCl, 0.6M NaCl with 0.5M H2SO4 and 0.6M NaCl with 1.0M H2SO4). The corroded surfaces were examined through SEM and EDS mapping. The corrosion mechanism of this novel promising alloy was discussed in a systematic and professional way. The corrosion results presented in this paper is also promising since the corrosion resistance is competing and even better that the commercial stainless steels as reference alloy (besides the already know promising deformability and mechanical properties of the same alloy). This means, with the current presented results in this paper, a new promising alloy with excellent mechanical and corrosion properties is explored for further and future investigations and potential applications by the concerned researchers and engineers in the field = high potential of citation.
Regarding the EBSD to check the texture of the alloy, we did not have such facility in our university or area. instead of this, we perform other tests which we believe it give us enough information to evaluate the microstructure, especially it is not the main originality of the paper. For example, if there is clear texture in the alloy after cold rolling, this will surely appear in the XRD results by having pronounced relative peaks intensity change, which is not the case in our alloy, therefore, expecting that there is a texture will affect the corrosion performance is scientifically has no evidence. So, performing the unavailable EBSD tests is not urgent in our case. Furthermore, as we explained in the paper, the corrosion behavior change is related mainly to phases change and elemental dissolution not texturing.
Thanks for your guidance and I hope to cooperate with you in the near future.
Best Regards

Reviewer 3 Report
Thank you for the answers. However, I noticed that Authors wrote that they changed images in Figures 10 and 11 for the ones with better quality, however, they are not changed in the revised manuscript. Therefore I ask Authors to change these two figures.
Author Response
Dear Prof. Reviewer
We thank you very much for your concern about our paper (Ref. No.: materials-1867457) titled “Corrosion behavior of cold rolled and solution treated Fe36Mn20Ni20Cr16Al5Si3 HEA in different acidic solutions" and appreciate your valuable comments on our work. Kindly, please find our reply on your comments in the following points.
Comment #1: Thank you for the answers. However, I noticed that Authors wrote that they changed images in Figures 10 and 11 for the ones with better quality, however, they are not changed in the revised manuscript. Therefore I ask Authors to change these two figures.
Authors Reply
The resolutions of Figures 10 and 11 were improved and inserted in the modified manuscript.
Thanks for your guidance and I hope to cooperate with you in the near future.
Best Regards
